# Dismantling the Papez circuit for memory in rats

Seralynne D Vann*

School of Psychology, Cardiff University, Cardiff, United Kingdom

**Abstract** Over the last 50 years, anatomical models of memory have repeatedly highlighted the hippocampal inputs to the mammillary bodies via the postcommissural fornix. Such models downplay other projections to the mammillary bodies, leaving them largely ignored. The present study challenged this dominant view by removing, in rats, the two principal inputs reaching the mammillary bodies: the postcommissural fornix from the hippocampal formation and Gudden's ventral tegmental nucleus. The principal mammillary body output pathway, the mammillothalamic tract, was disconnected in a third group. Only mammillothalamic tract and Gudden's ventral tegmental nucleus lesions impaired behavioral tests of spatial working memory and, in particular, disrupted the use of extramaze spatial landmarks. The same lesions also produced widespread reductions in immediate-early gene (c-*fos*) expression in a network of memory-related regions, not seen after postcommissural fornix lesions. These findings are inconsistent with previous models of mammillary body function (those dominated by hippocampal inputs) and herald a new understanding of why specific diencephalic structures are vital for memory.

## Introduction

Since the first neuroanatomical circuits of emotion (e.g., *Papez, 1937*) and memory (e.g., *Barbizet, 1963*; *Delay and Brion, 1969*), the perceived contributions of the mammillary bodies have been dominated by their direct inputs from the hippocampal formation (via the fornix). Consequently, the mammillary bodies are principally seen as a relay of hippocampal projections to the anterior thalamic nuclei and, from there, to the cingulate (*Barbizet, 1963*; *Delay and Brion, 1969*) and prefrontal (*Warrington and Weiskrantz, 1982*) cortices. The notion that the mammillary bodies form part of such an 'extended hippocampal system' (*Aggleton and Brown, 1999*) is seemingly supported by the unidirectional nature of the hippocampal projections to the mammillary bodies (*Aggleton et al., 2005*), along with clinical evidence of the importance for memory of the mammillary body projections to the thalamus, via the mammillothalamic tract (MTT; *Carlesimo et al., 2007*; *Van der Werf et al., 2000*; *Vann and Aggleton, 2003*). However, these standard models suffer two major shortcomings. First, they provide no real function for the mammillary bodies and second, they ignore other mammillary body afferents. The mammillary bodies receive dense inputs from Gudden's tegmental nuclei (e.g., *Takeuchi et al., 1985*), making it possible that these midbrain inputs are functionally important for memory (*Vann, 2010*).

Three groups of rats contrasted the standard model with an alternative model of mammillary body function based on its tegmental inputs. Rats with lesions blocking either of the two main mammillary body afferents (hippocampus formation or ventral tegmental nucleus of Gudden) were tested alongside rats with lesions of the main mammillary body efferent pathway, the mammillothalamic tract. Only by comparing these separate lesions within the same study is it possible to measure their relative contributions. In one group the descending postcommissural fornix (PCF) was lesioned, a procedure that disconnects the hippocampal formation projections to the mammillary bodies whilst leaving other hippocampal connections intact, including those with the septum, striatum and anterior thalamic

*For correspondence: vannsd@cardiff.ac.uk

**Competing interests:** The author declares that no competing interests exist.

**Reviewing editor**: Howard Eichenbaum, Boston University, United States

**eLife Digest** The hippocampus is a seahorse-shaped structure in the brain and its role in memory has been recognized since the 1950s. However, much less is known about two small structures called the mammillary bodies that are found near the hippocampus. These bodies are part of the limbic system—a network of brain regions that also includes the hippocampus and the amygdala—and this system is known to be involved in the regulation of emotion and the formation of long-term memories.

In 1937, James Papez injected rabies virus into the hippocampus and, by tracing its movement through the brain, identified a distinct circuit within the limbic system. This circuit, which is today known as Papez' circuit, consists of projections from the hippocampal formation to the mammillary bodies, and from the mammillary bodies on to another region called the anterior thalamus. From here, projections form a loop via several other regions back to the hippocampus.

It is widely thought that the mammillary bodies are required for memory formation due to their role in relaying projections from the hippocampus. However, the mammillary bodies also receive projections from other regions, including Gudden's ventral tegmental nucleus, and it is possible that these could contribute to the role of the mammillary bodies in memory.

To distinguish between these possibilities, Seralynne Vann compared the performance of three groups of lesioned rats in tests of spatial short-term memory. The first group had lesions of the hippocampal inputs to the mammillary bodies; the second had lesions of the ventral tegmental inputs to the mammillary bodies; and the third group had lesions of the mammillary body outputs to the thalamus. Vann found that the third group was impaired in the memory tasks, consistent with the idea that outputs sent from the mammillary bodies to the thalamus are required for memory formation. Surprisingly, however, blocking signals sent from the hippocampal formation to the mammillary bodies had little impact on the formation of memories, whereas blocking inputs from Gudden's ventral tegmental nucleus led to significant impairments in memory.

By revealing that limbic midbrain inputs to the mammillary bodies have an essential role in the formation of memories, these new results challenge dogma in the field, and highlight the importance of looking beyond the hippocampus when considering memory.

nuclei (*Vann et al., 2011*). A second group received neurotoxic lesions of Gudden's ventral tegmental nucleus (VTNg). Although the lateral mammillary nucleus receives dense inputs from Gudden's dorsal tegmental nucleus, the VTNg was targeted because of its inputs to the medial mammillary nucleus. Previous clinical and behavioral evidence highlights the particular importance of the medial mammillary nucleus for memory (*Vann, 2005*, *2010*, *2011*; *Vann and Albasser, 2009*). A third group received mammillothalamic tract lesions while a fourth group underwent control surgery. Rats were tested on three behavioral tasks, the first two of which are sensitive to hippocampal damage: T-maze alternation, a working memory task in the radial-arm maze, and a geometric discrimination task (*Aggleton et al., 2009*; *Vann, 2011*).

In addition to impairing spatial memory, MTT lesions also result in widespread neural hypoactivity, as measured by expression of the immediate-early gene c-*fos* (*Vann and Albasser, 2009*). To determine whether these changes reflect the indirect loss of hippocampal afferents or the disconnection of tegmental pathways, tissue from all four groups of animals was processed immunohistochemically for the expression of c-*fos*.

## Results

### Histological analysis

A stringent lesion criterion was adopted for final inclusion in the study and, as a consequence, the final groups comprised seven Gudden's ventral tegmental nuclei lesion animals (VTNx), eight mammillothalamic tract lesion animals (MTTx), seven postcommissural fornix lesions (PCFx) and eight surgical controls (Sham).

#### Gudden's ventral tegmental nucleus lesion

The VTNg lesions, which were assessed using Nissl and acetylcholinesterase stained sections (*Figure 1A,B*), produced a clear loss of the large, distinctive cells in the ventral tegmental nucleus.

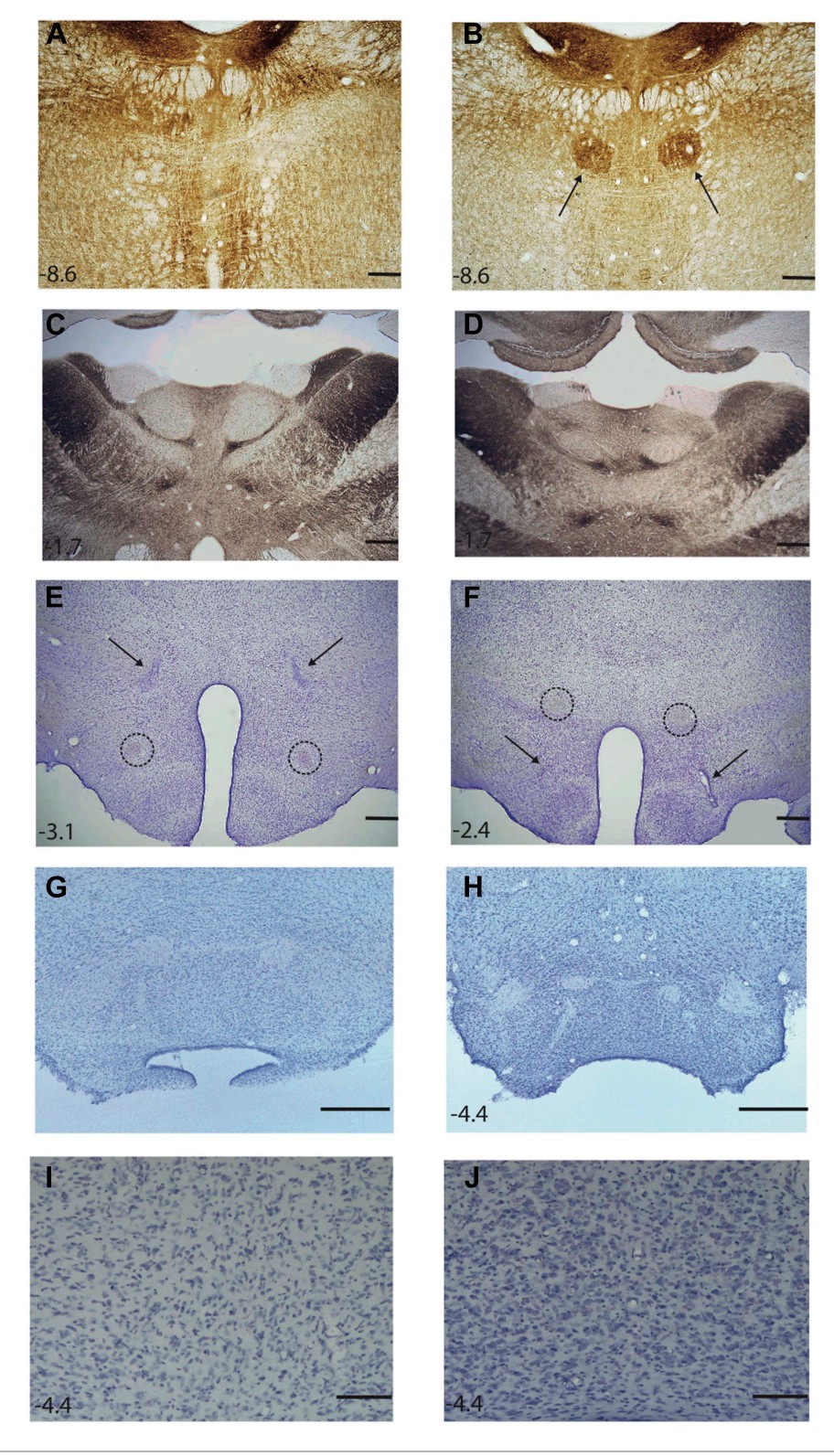

**Figure 1**. Photomicrographs depicting typical lesions. (**A**) Acetylcholinesterase-stained section showing ventral tegmental nucleus of Gudden lesion. (**B**) Acetylcholinesterase-stained section showing the ventral tegmental nucleus of Gudden (indicated by the arrows) in an intact animal. (**C**) Acetylocholinesterase staining in the
*Figure 1. Continued on next page*

*Figure 1. Continued*

anteroventral thalamic nuclei in a ventral tegmental nucleus lesion rat; and (**D**) a surgical control rat; (**E**) Nissl-stained section showing mammillothalamic tract lesion, the arrows indicate the lesion. The intact postcommissural fornix is highlighted within the circles. (**F**) Nissl-stained section showing postcommissural fornix lesion, the arrows indicate the lesion and the intact mammillothalamic tract is highlighted within the circles. (**G**) Mammillary bodies from a surgical control; (**H**) Mammillary body atrophy following postcommissural fornix lesion. (**I**) A higher magnification of (**G**) showing Nissl-stained cells in the medial mammillary nucleus of a surgical control shown. (**J**) A higher magnification of (**H**) showing increased cell packing in the medial mammillary nucleus following a post commissural fornix lesion. Numbers indicate distance in millimeters from bregma. Scale bar: (**A**–**H**), 500 µm; (**I**–**J**), 100 µm.

In all animals there was very limited damage to the medial-most part of the oral part of the reticular pontine nucleus; this damage was always unilateral and on the contralateral side of the brain to that in which the needle was inserted to make the lesion. There did not appear to be any loss of raphe neurons at any level other than that of the VTNg. At just this level there was a restricted loss of cells in the most dorsal part of the median raphe nuclei, that is, limited to just that part directly adjacent to the VTNg. There was no evidence that the lesions extended into either Gudden's dorsal tegmental nuclei or the laterodorsal tegmental nuclei, both of which stained intensely for acetylcholinesterase.

The integrity of the laterodorsal tegmental nucleus is important, being the sole source of anterior thalamic cholinergic innervations and a major source of interpeduncular cholinergic inputs (**Satoh and Fibiger, 1986**; **Sikes and Vogt, 1987**). The observation that the laterodorsal tegmental nuclei, and subsequently their cholinergic projections, were intact was confirmed by assessing the level of acetylcholinesterase staining in the anteroventral thalamic nucleus (**Figure 1C,D**) and interpeduncular nucleus. The mean gray values ± standard deviation for the anteroventral thalamic nuclei were: VTNx, 75.8 ± 9.05; Sham, 79.3 ± 4.17, $t < 1$. The interpeduncular nucleus values (rostral part) were: VTNx, 73.4 ± 10.9; Sham, 75.8 ± 12.8 ($t < 1$).

The PCFx and MTTx lesions were assessed using Nissl stained sections (**Figure 1E,F**); any cases with unilateral or bilateral sparing were removed from further analyses. It has been repeatedly shown that fornix section leads to shrinkage of the mammillary bodies (due to loss of white matter). Mammillary body area measurements were, therefore, made to determine the impact of the PCFx lesions. The mean mammillary body area (±SEM) for the PCFx group was 1.34 mm² ± 0.031, making them significantly smaller than the Sham group measurements of 1.79 mm² ± 0.058 ($t(13) = 6.44$; $p<0.001$; **Figure 1G,H**). Furthermore, the reduced mammillary body volumes of the present PCFx lesion cohort did not differ from a previous cohort where retrograde tracers had been used to confirm the postcommissural fornix disconnection ($t(14)=1.67$; $p>0.1$) or from rats with complete fornix lesions ($t < 1$) (**Vann et al., 2011**). These findings demonstrate quantitative changes in mammillary body status, consistent with a complete tract section. The mammillary bodies were also assessed in the MTTx and VTNx lesion groups (MTTx = 1.38 mm² ± 0.045; VTNx = 1.50 mm² ± 0.026). When considering the mammillary body cross-sectional area of all four groups in the study there was a main effect of group ($F_{(3,27)} = 22.3$, $p<0.001$). Post-hoc comparisons revealed the mammillary bodies to be larger in the Sham group than all lesion groups (all $p<0.001$) while none of the lesion groups differed significantly from each other (all $p>0.05$). To determine whether the reduced medial mammillary bodies volumes represented loss of white matter, as has been previously described (**Loftus et al., 2000**), cell counts were made (cell counts per mm² ± SEM: Sham = 3530.4 ± 77.6; PCFx = 4054.7 ± 96.6; MTTx = 3405.4 ± 89.8; VTNx = 3361.1 ± 134,3). There was an overall group effect ($F_{(3,27)} = 9.96$, $p<0.001$) which reflected a significant increase in the cell packing in the PCFx lesion group compared to all three other groups (Sham, $p<0.01$; MTTx, $p<0.005$; VTNx, $p<0.001$) there was no difference in cell packing in any of the other groups (all comparisons, $p>0.5$). This is consistent with the reduced mammillary body volume in the PCFx group reflecting loss of white matter and not an overall change in neuronal numbers (**Figure 1I,J**). In contrast, lack of difference in cell packing in the MTTx and VTNx groups, combined with reduced cross-sectional area, would be consistent with some medial mammillary body cell loss in these groups. It is important to note that these counts are not stereological so cannot be used to determine total cell counts for the mammillary bodies.

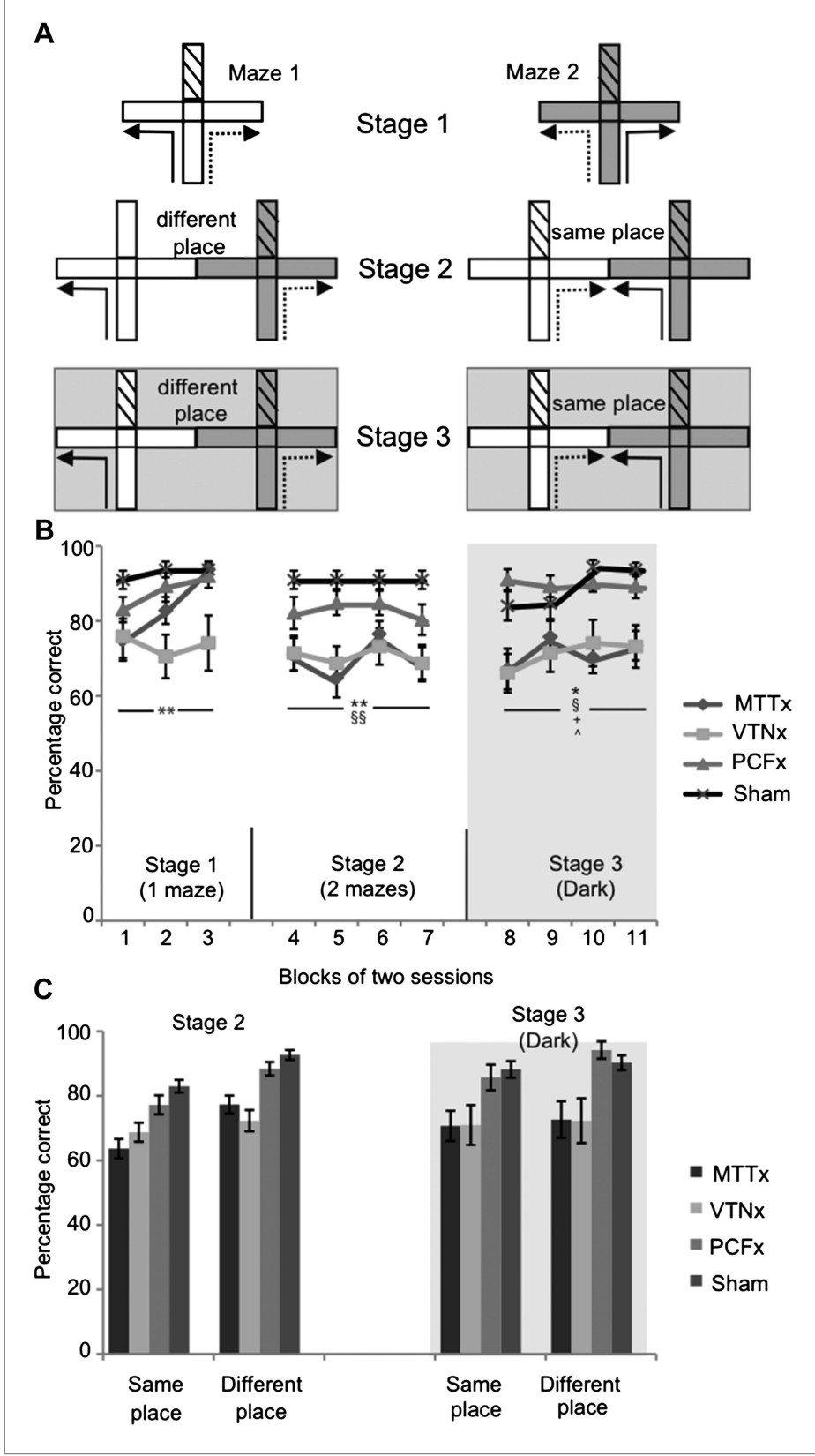

**Figure 2**. T-maze alternation. (**A**). Experimental design for the three stages: solid lines indicate the forced sample phase while dashed lines indicate the correct response in the choice phase. By placing a barrier at the

*Figure 2. Continued on next page*

*Figure 2. Continued*

entrance of the arm access to an arm could be blocked (effectively turning the cross-maze into a T-maze configuration; this is illustrated by hatching). Initial training on the task (Stage 1) permitted the use of multiple strategies supporting alternation, that is allocentric, intra-maze, idiothetic, direction alternation (with reference to a known bearing). The task was then systematically modified in order to prevent the use of intra-maze cues (Stage 2) or the use of intra-maze and distal allocentric cues (Stage 3). These manipulations included using two mazes instead of one (Stage 2 and 3) or running in the dark, as illustrated with the gray background (Stage 3); (**B**) Mean percentage of correct choices (±SEM) for all three stages; (**C**) percentage of correct choices for the 'same place' and 'different place' trials in Stages 2 and 3. The vertical lines depict the standard error of the mean. Abbreviations: p; Sham, surgical control; *Significant difference between VTNx and Sham group (p<0.05); **Significant difference between VTNx and Sham group (p<0.01); §Significant difference between MTTx and Sham group (p<0.05); §§Significant difference between MTTx and Sham group (p<0.01); +Significant difference between PCFx and MTTx (p<0.05); ^Significant difference between PCFx and VTNx (p<0.05).

## Behavioral results

### Experiment 1: reinforced spatial alternation in the T-maze

Rats can use multiple cue-types to perform spatial alternation tasks (e.g., *Douglas, 1966*; *Dudchenko, 2001*), including visual allocentric cues, intra-maze cues (e.g., odor trails), directional cues (i.e., bearing), and egocentric cues (i.e., body turns). The alternation training, therefore, consisted of three sequential stages (*Figure 2A*) during which the use of specific cue-types was permitted or excluded.

In Stage 1, both the sample and choice run were carried out in the same maze (*Figure 2A*). Both mazes were used equally, each maze alternating between sessions. For these trials, all cue types could support performance (i.e., visual allocentric, intra-maze, direction, and egocentric). There was an overall effect of lesion ($F_{(3,27)}$ = 4.93, p=0.007; *Figure 2B*) and an overall effect of training block ($F_{(1.44,38.9)}$ = 3.50, p=0.037) but no lesion group x block interaction ($F_{(4.33,38.9)}$ = 1.52, p>0.1). Post-hoc analyses revealed the performance of the VTNx group to be significantly worse than the Sham group (p=0.005). None of the other group comparisons was significant (all p>0.05).

In Stage 2, the sample and choice runs were conducted in adjacent mazes to prevent the use of intra-maze cues while visual allocentric, direction (bearing) and egocentric cues (but see below) remained available. There was a significant, overall group effect on Stage 2 ($F_{(3,27)}$ = 7.12, p=0.001; *Figure 2B*) but no effect of block ($F$ < 1) and no group x block interaction ($F$ < 1). The group effect reflected the lower scores of both the MTTx (p=0.004) and VTNx (p=0.006) groups compared to the Sham rats. These results suggest that the lack of impairment in the MTTx group in Stage 1 reflected their use of intra-maze cues, such as odor trails.

The two adjacent mazes created two distinct trial types (*Figure 2A*). For half of the trials ('different place') the correct choice took the rat further away from the arm used in the sample run (*Figure 2A*). For the remaining trials ('same place', *Figure 2A*), the rats were rewarded for choosing the arm leading to a goal located very close to the location of the food in the sample run. Overall, same place trials were more difficult than different place trials ($F_{(1,27)}$ = 25.4, p<0.001) but no there was trial type x group interaction ($F_{(3,27)}$ = 1.23, p>0.3; *Figure 2C*).

Stage 3 was identical to Stage 2, except it was run in the dark to minimize the use of visual allocentric cues (in addition to those intra-maze cues removed by the use of two mazes). While both direction cues and egocentric cues were available during this stage, previous studies have shown that rats find it very difficult to use egocentric cues (e.g., body turns) for delayed alternation (*Baird et al., 2004*; *Futter and Aggleton, 2006*; *Pothuizen et al., 2008*) making direction the more likely cue. An overall analysis of Stage 3 revealed group differences ($F_{(3,27)}$ = 6.28, p=0.002) but no effect of block ($F_{(3,81)}$ = 2.19, p>0.3) and no block x group interaction ($F_{(9,81)}$ = 1.08, p>0.3; *Figure 1B*). Post-hoc analyses revealed that the performances of the VTNx and MTTx groups were significantly worse than those of both the Sham group (VTNx, p=0.025; MTTx, p=0.019) and the PCFx group (VTNx, p=0.029; MTTx, p=0.023). The lack of a trial type effect ('same' vs 'different' $F_{(1,27)}$ = 3.60, p>0.05) is to be expected if being in the dark removed the visual cues that had previously helped the rats to discriminate food well locations (*Figure 2C*).

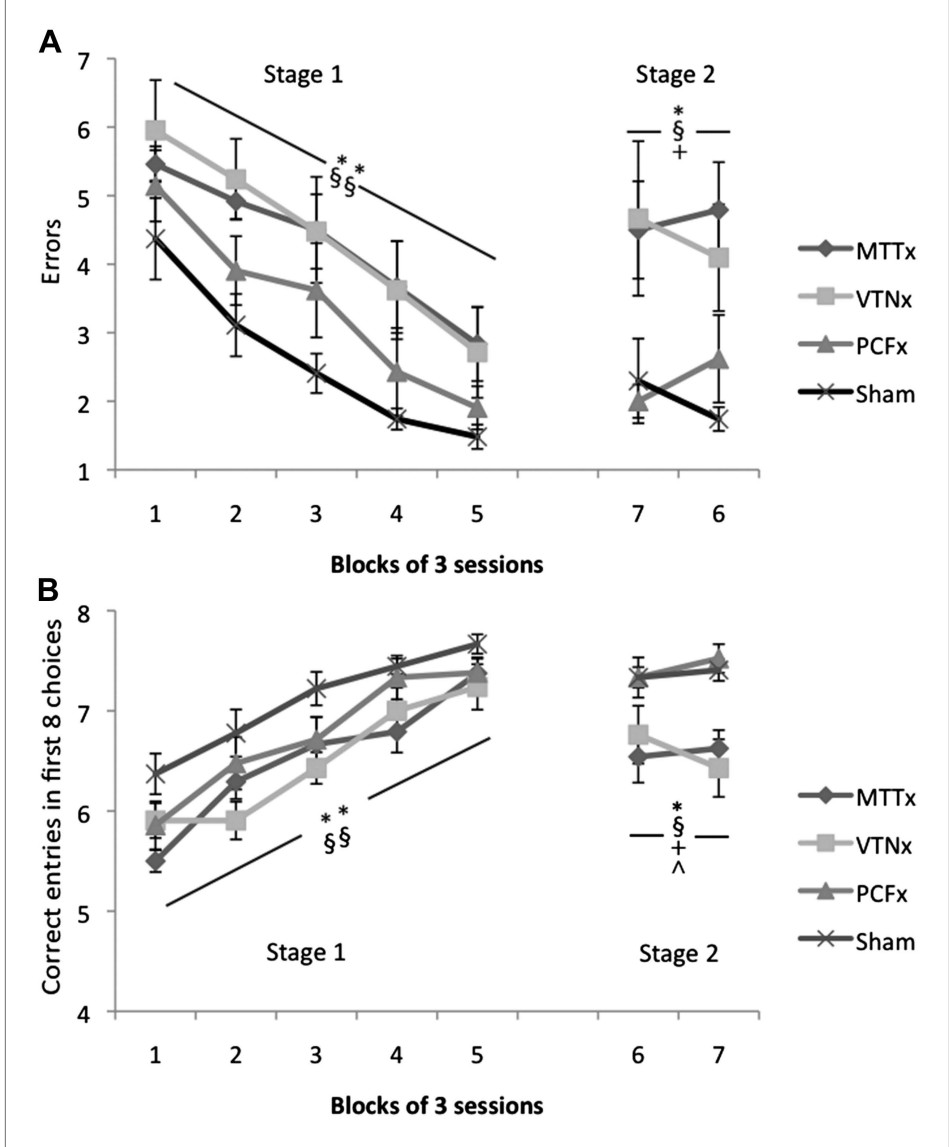

**Figure 3**. Radial-arm maze task. (**A**) Mean number of errors (±SEM). First five blocks represent acquisition of the tasks and the final two blocks include rotation of the maze; (**B**) Mean number of correct entries in first eight arm choices (±SEM) during acquisition (first five blocks) and rotation (final two blocks); *Significant difference between VTNx and Sham group (p<0.05); **Significant difference between VTNx and Sham group (p<0.01); §Significant difference between MTTx and Sham group (p<0.05); §§Significant difference between MTTx and Sham group (p<0.01); +Significant difference between PCFx and MTTx (p<0.05); ^Significant difference between PCFx and VTNx (p<0.05).

## Experiment 2: radial-arm maze (acquisition and rotation)

Rat were tested on the working memory version of the radial-arm maze task (*Olton et al., 1978*) where the animals' optimal strategy is to retrieve the reward pellets from all eight arms without re-entering any previously entered arms. Both total errors and number of correct entries in the first eight choices were analyzed. For the acquisition phase (Stage 1) there was a significant effect of group using both errors ($F_{(3,27)}$ = 6.20, p=0.002; *Figure 3A*) and correct entries ($F_{(3,27)}$ = 6.54, p=0.002; *Figure 3B*); in both instances post-hoc comparisons showed this group effect to be due to significant differences between the Sham group and both the MTTx (errors, p=0.007; entries, p=0.005) and VTNx groups (errors, p=0.005; entries, p=0.004). There was also a main effect of training block using both measures (errors: $F_{(4,108)}$ = 28.3, p<0.001; entries: $F_{(4,108)}$ = 47.0, p<0.001) but no group x block interaction (errors and entries, both $F$ < 1) reflecting the improvement in all of the groups' performance.

For Stage 2, the maze was rotated after each animal had made its first four arm choices; this manipulation taxed the use of extra-maze cues by making intra-maze information unreliable. Following this manipulation the groups significantly differed in terms of number of errors made ($F_{(3,27)}$ = 6.35, p=0.002; *Figure 3A*) and correct entries in the first eight choices ($F_{(3,27)}$ = 6.80, p=0.001; *Figure 3B*). Post-hoc comparisons revealed this group effect to be due to differences between the Sham group and both the MTTx (errors, p=0.007; entries, p=0.015) and VTNx groups (errors, p=0.023; entries, p=0.022). In addition, there were significant differences between the PCFx and MTTx scores using both error and correct entry values (errors, p=0.030; entries, p=0.014) and between the PCFx and VTNx groups using correct entries (p=0.020). There was no effect of block using either measure (both $F < 1$) nor was there a group x block interaction (errors: $F < 1$; entries: $F_{(3,27)}$ = 1.20, p>0.3).

## Experiment 3: geometric discrimination in the water-maze

Rats were required to use visual cues, that is, the configuration of long and short walls in a rectangle, to navigate to a hidden platform in a water-maze (*Figure 4A*). All rats received one session of pre-training in the circular pool (without the rectangular insert) with a landmark attached to the platform. There was no difference between any of the groups' escape latencies for this pre-training session in the circular pool ($F_{(3,27)}$ = 1.48, p>0.2; means ± SEM, MTTx = 26.1 ± 3.1 VTNx = 31.8 ± 3.6; PCFx = 23.6 ± 2.2; Sham = 29.7 ± 2.5). Rats were subsequently trained with a rectangular insert in the pool, with no landmark attached to the platforms; they were required to learn the two geometrically identical correct corners in order to escape from the pool (*Figure 4A*). All groups of rats were able to learn this task as demonstrated by a lack of group effect ($F_{(3,27)}$ = 2.38, p>0.05), a significant effect of training block ($F_{(3,81)}$ = 62.8, p<0.001; *Figure 4B*) and no group x block interaction ($F < 1$). There were no group differences in swim velocity during acquisition ($F_{(3,27)}$ = 2.44, p>0.05) nor was there a group x training block interaction ($F_{(10.1,91.3)}$ = 1.59, p>0.1). A single probe session (Session 13) was conducted upon completion of the above training where the rats were allowed to swim in the rectangle for 60s with no escape platform. There was no effect of group ($F < 1$) but all groups had clearly learnt to discriminate the geometry of the rectangle as reflected by a significantly longer time spent in the correct corners compared to incorrect corners ($F_{(1,27)}$ = 207.8, p<0.001; *Figure 4C*) and lack of group by corner interaction ($F_{(3,27)}$ = 2.43, p>0.05; *Figure 4C*).

# Distal changes as measured by the immediate-early gene c-*fos*

## Experiment 4: c-*fos* expression following a radial-arm maze task

### Behavior

As two of the lesion groups showed clear spatial memory deficits, rats were tested on a forced-choice version of the radial-arm maze task so sensori-motor behavior could be controlled across groups and reward contingencies could be matched. On the final test day the rats were tested on the same task but in a novel room. This manipulation raises c-Fos levels in a network of hippocampal-related regions (see *Jenkins et al., 2002*) while rapid, new spatial learning appears dependent on MTT fibers (*Vann and Aggleton, 2003*). On the final test day, when animals were tested in the novel room, all animals completed approximately four trials (a trial is completed when all eight arms have been visited) during the 20 min test session (trials completed: VTNx= 4.0 ± 0.0; MTTx = 4.0 ± 0.0; PCFx = 3.9 ± 0.1; Sham = 4.0 ± 0.1), with no group difference in trial numbers ($F_{(3,27)}$ = 1.85, p>0.1).

### c-Fos counts

c-Fos counts were made in a number of memory-related areas, including the hippocampal formation, retrosplenial cortex and prefrontal cortex. Counts were also taken from additional brain areas that may be innervated by the VTNg or considered control areas.

### Retrosplenial cortex

There was a significant group effect ($F_{(3,26)}$ = 11.97, p<0.001; *Figure 5A*) as the Sham group had more Fos-positive cells compared to both the MTTx (p<0.001) and VTNx groups (p=0.006). There were also significantly more Fos-positive cells in the PCFx lesion group relative to both the MTTx (p=0.001) and VTNx lesion (p=0.026) groups (*Figure 6*).

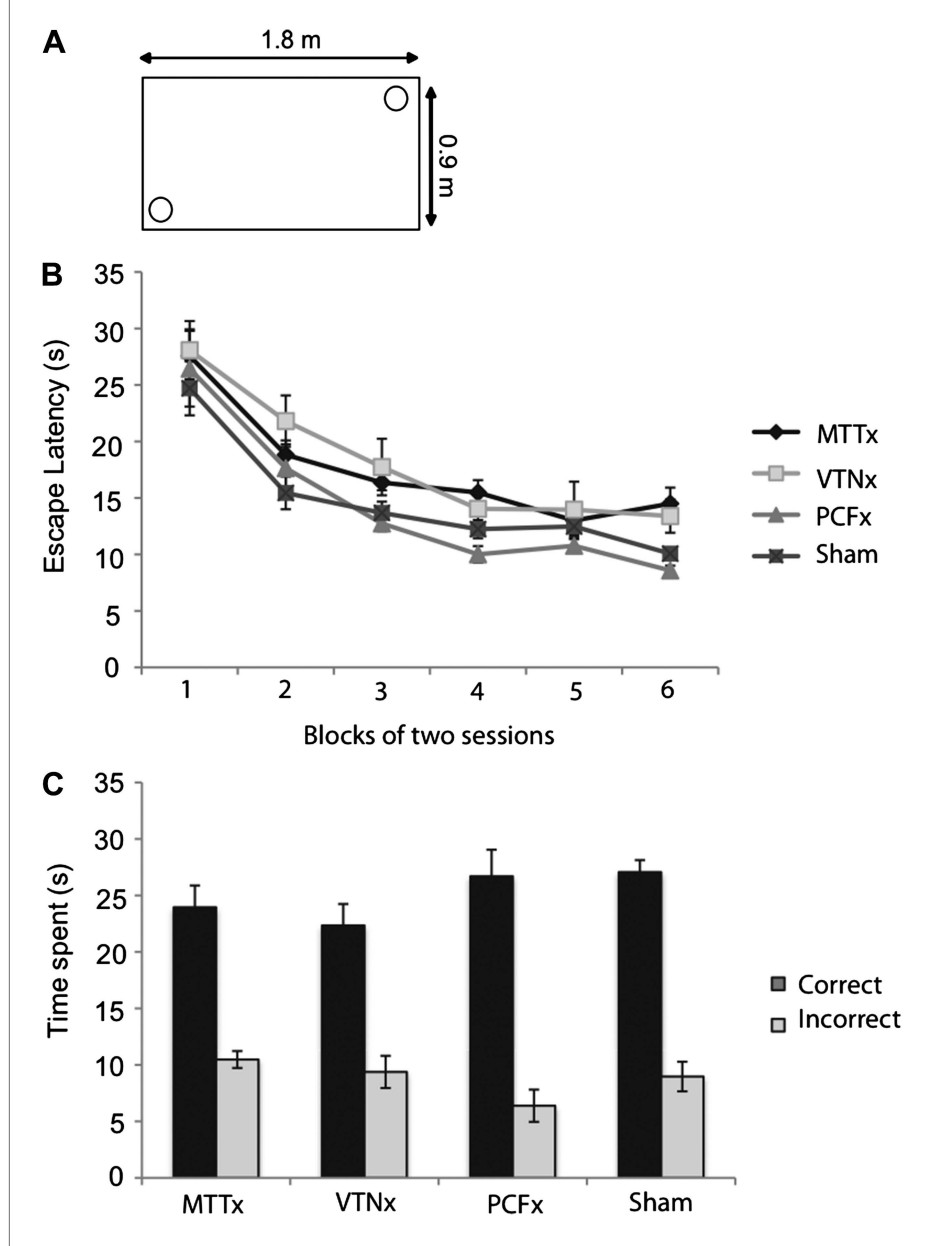

**Figure 4**. Geometric discrimination in the water-maze. (**A**) A schematic of the platform positions for the half of the animals (i.e., short wall to the right of long wall). (**B**) Mean escape latencies (±SEM) for task acquisition. (**C**) Probe performance. The platforms are removed from the pool and the rat is allowed to swim for 60s. The mean times spent (±SEM) in the correct corners and the incorrect corners are presented.

## Hippocampus

Hippocampal Fos-positive cell counts included all subfields (dentate gyrus, CA3 and CA1) and were taken at a level where both intermediate and temporal hippocampus are present. Fos-positive cells were restricted to the stratum pyramidale of the CA fields and the stratum granulosum of the dentate gyrus. There was a main group effect ($F_{(3,26)}$ = 5.52, p=0.05; **Figure 5A**) as c-Fos counts in the Sham group were significantly higher than both the MTTx group (p=0.029) and VTNx groups (p=0.009). The dentate gyrus (Sham = 31.0 ± 2.93; MTTx = 24.6 ± 4.91; VTNx = 21.0 ± 3.01; PCFx = 29.0 ± 4.30), CA3 (Sham = 23.6 ± 3.21; MTTx = 15.0 ± 2.72; VTNx = 18.1 ± 3.31; PCFx = 24.1 ± 2.82), and CA1 (Sham = 178.3 ± 14.0; MTTx = 110.7 ± 10.3; VTNx = 106.8 ± 14.8; PCFx = 144.2 ± 6.26), were also analyzed separately. These analyses revealed that the lesion-induced changes in hippocampual

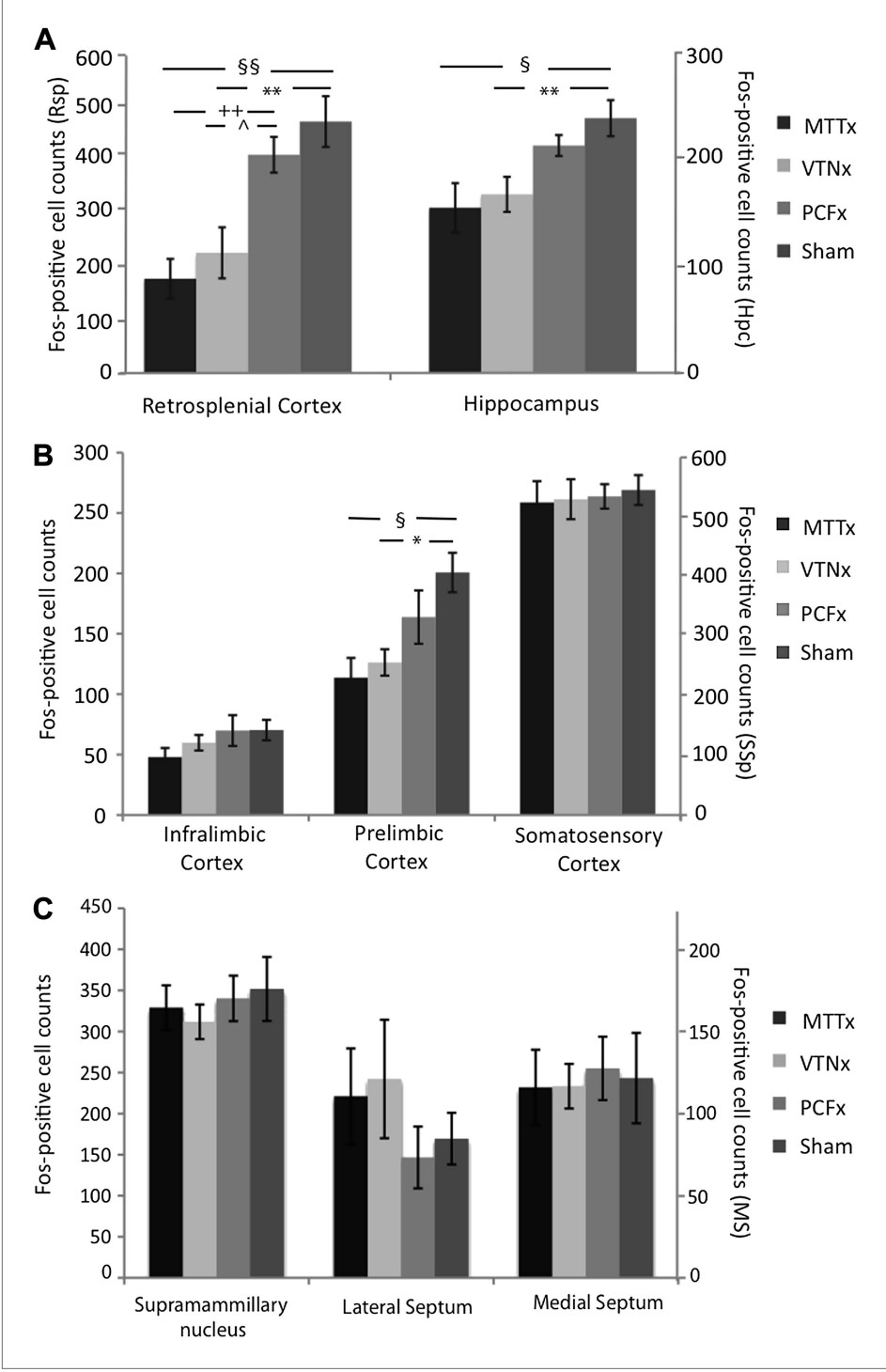

**Figure 5**. Mean c-Fos-positive cell counts (±SEM). (**A**) Retrosplenial cortex and hippocampal formation (dentate gyrus, CA1, CA3); (**B**) Frontal cortices (infralimbic cortex and prelimbic cortex) and somatosensory cortex. (**C**) Supramammillary nuclei, lateral and medial septum. *Significant difference between VTNx and Sham group (p<0.05); **Significant difference between VTNx and Sham group (p<0.01); §Significant difference between MTTx and Sham group (p<0.05); §§Significant difference between MTTx and Sham group (p<0.01); ++Significant differences between PCFx (p<0.01); ^Significant difference between PCFx and VTNx (p<0.05).

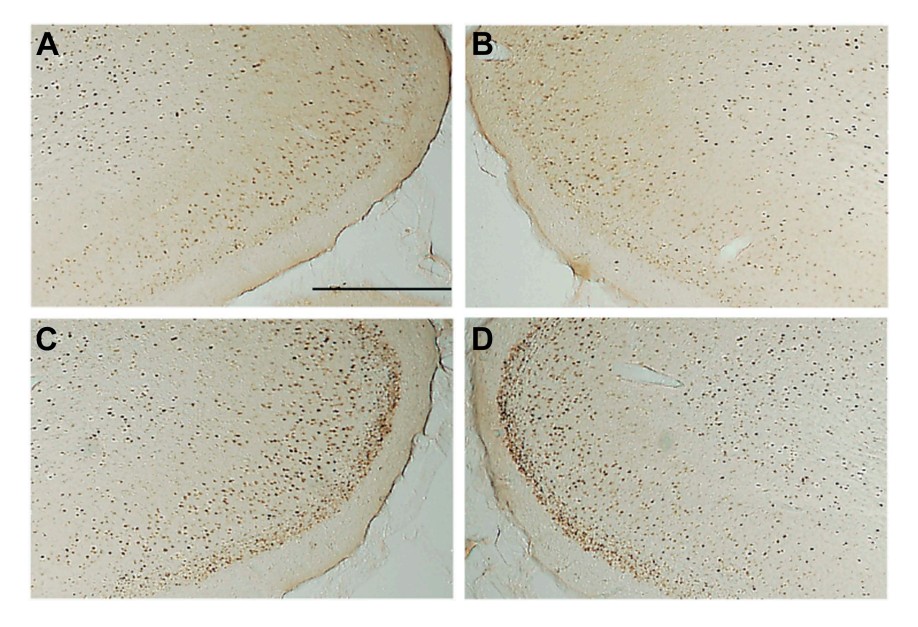

**Figure 6**. Photomicrographs of Fos-positive nuclei in the retrosplenial cortex in the four groups: (**A**) mammillothalamic tract lesion (**B**) ventral tegmental nucleus of Gudden lesion; (**C**) descending postcommissural fornix lesion group (**D**) surgical control. The darkly stained cells are Fos-postitive neurons. The loss of distinctive c-Fos staining in layer two can been seen in the mammillothalamic tract lesion (**A**) and ventral tegmental nucleus of Gudden (**B**). Scale bar, 500 µm.

c-*fos* expression appeared to be largely driven by reductions in CA1 as this was the only subfield where there was a significant group difference (dentate gyrus: $F < 1$; CA3: $F_{(3,26)} = 2.01$, $p>0.4$; CA1: $F_{(3,26)} = 8.07$, $p<0.005$). The group difference in CA1 arose due to fewer Fos-positive cell counts in both the MTTx and VTNx groups compared to the Sham group (both $p<0.005$). None of the other group comparisons were significant (all $p>0.1$).

## Frontal regions

There was no group effect when analysing the infralimbic cortex ($F_{(3,26)} = 1.23$, $p>0.3$; *Figure 5B*). In contrast, analyses of the prelimbic cortex revealed an overall group effect ($F_{(3,26)} = 5.32$, $p=0.006$; *Figure 5B*) as significantly fewer c-Fos positive cells were found in the MTTx ( $p=0.041$) and VTNx ($p=0.017$) groups than in the Sham group.

## Supramammillary nucleus, septum, and somatosensory cortex

There was no group effect for somatosensory cortex ($F < 1$; *Figure 5C*), supramammillary nucleus, lateral or medial septum (all $F < 1$; *Figure 5C*).

## Discussion

Although the mammillary bodies (MBs) have a very longstanding association with memory (*Gudden, 1896*), remarkably little is known about their specific contributions. Models of MB function have inevitably focused on their dense hippocampal inputs (*Aggleton and Brown, 1999*) so that the MBs are principally thought to relay hippocampal information to the anterior thalamic nuclei (via the mammillothalamic tract), and then beyond (e.g., *Barbizet, 1963*; *Delay and Brion, 1969*). However, the MBs have heavy, reciprocal connections with Gudden's tegmental nuclei whose functional significance has been largely overlooked. The present study contrasted the relative contributions of these two, principal MB inputs for (i) spatial memory and (ii) control of the expression of the immediate early gene c-*fos* in distal limbic sites. The second goal arose because mammillothalamic tract lesions cause c-Fos hypoactivity in numerous sites including the hippocampus, retrosplenial cortex, and prelimbic cortex (*Vann and Albasser, 2009*). The present study determined whether it is the loss of hippocampal or tegmental

inputs that is the main cause of this neural hypoactivity. For these joint reasons, rats with either lesions of the descending postcommissural fornix (PCFx), which carries the hippocampal formation projections, or lesions of Gudden's ventral tegmental nucleus (VTNx) were tested alongside mammillothalamic tract lesion rats (MTTx), that is, the major efferent tract by which the MBs have their principal effects (*Vann and Aggleton, 2003*).

The present study focused on the ventral tegmental nucleus of Gudden as it is reciprocally connected with the medial mammillary nucleus while the dorsal tegmental nucleus connects with the lateral mammillary nucleus. Although the lateral mammillary nucleus forms part of the head-direction network (e.g., *Vann and Aggleton, 2004*; *Hopkins, 2005*), it is the loss of the medial mammillary system that appears principally responsible for the spatial memory impairments associated with complete lesions of this structure (*Vann, 2005*, *2010*, *2011*; *Vann and Albasser, 2009*). Furthermore, it is the medial mammillary nucleus that is always atrophied in the amnesic Korsakoff's syndrome (*Victor et al., 1989*; *Kopelman, 1995*).

In contrast to the predictions of traditional models of MB function (e.g., *Delay and Brion, 1969*; *Gaffan, 1992*; *Aggleton and Brown, 1999*), only the MTTx and VTNx groups were impaired on the behavioral tasks relative to the Sham groups. At no stage did these two lesion groups significantly differ from one another. The only comparison where just one of these two groups was impaired was for Stage 1 of T-maze alternation where the VTNx, but not MTTx, group made significantly more errors than the Sham group. This apparent exception makes sense when the results from Stage 2 are considered. In Stage 1 of the T-maze alternation task both the sample and test runs were in the same maze; in Stage 2 the sample and test runs were in adjacent mazes to prevent the use of intra-maze cues. The performances of the VTNx and MTTx groups in Stage 2 were equivalent, with both groups impaired relative to the Sham group, suggesting the MTTx group had been using intra-maze cues to solve Stage 1, so masking their spatial memory impairments. The MTTx and VTNx groups were also equivalently impaired on the radial-arm maze task. Although all groups showed an improvement across training, it appears that in the VTNx and MTTx groups this reflected a greater reliance on the use of intra-maze cues given the disruptive effect of rotating the maze midway through the trial. Thus both behavioral tasks indicated that the MTTx and VTNx rats were less able to use allocentric cues and more reliant on intra-maze cues. In contrast, the PCFx animals were not impaired relative to the Shams on any of the behavioral tasks. Consequently, the PCFx animals performed significantly better than both the VTNx and MTTx groups on both the radial-arm maze and T-maze tasks despite producing MB shrinkage (resulting from loss of fiber inputs) equivalent to that found after complete fornix lesions. The present PCFx lesions were indistinguishable from those that had previously been demonstrated to completely disconnect the hippocampal formation-MB projections (*Vann et al., 2011*).

The pattern of behavioral impairments was paralleled by the c-Fos findings. Both the VTNx and MTTx groups had significantly fewer c-Fos positive cells in the hippocampus, retrosplenial cortex, and prelimbic cortex. Reduced levels of c-Fos have also been found in this same network of brain regions following anterior thalamic nuclei lesions (*Jenkins et al., 2002*). However, unlike the c-Fos reduction following anterior thalamic lesions, the current changes reflect 'indirect' effects, as the MTT and VTNg lesions do not result in direct de-afferentation of any of these sites. Although reduced activity in this memory-related network seems to be a consistent marker of damage to the medial diencephalon, in animal models and humans (*Joyce et al., 1994*; *Kapur, 1994*; *Reed et al., 2003*; *Savage et al., 2003*; *Caulo et al., 2005*; *Anzalone et al., 2010*), it is not known whether this reduced activity contributes to mnemonic deficits or whether it simply a symptom of the system no longer working effectively. Evidence for the former account comes learning and memory deficits found after infusions of antisense that block c-Fos activity (*He et al., 2002*; *Seoane et al., 2012*; *Katche et al., 2013*).

The present PCFx lesion findings were remarkably similar to those found previously (*Vann et al., 2011*), for example, they resulted in a 5% reduction in accuracy of T-maze alternation. The lack of effect of PCFx lesions is quite surprising and does raise the question of what does this pathway do. It is possible that the tasks used in the present study did not sufficiently tax this pathway and that greater task demands could result in lesion-induced impairments. In addition, it may be that these projections contribute to memory under normal conditions but the general lack of effects following PCFx lesions are due to redundancies in this pathway, given the direct projections from the hippocampal formation to the anterior thalamic nuclei. While complete fornix lesions disconnect the hippocampal formation from a number of sites, including their projections to both the MBs and anterior thalamic nuclei, selective lesions of the descending PCF will spare all but the MB projections. The hippocampal projections to

the MBs and anterior thalamic nuclei arise from different cell populations (*Wright et al., 2010*) so it is not clear whether the same information is being conveyed (*Tsanov et al., 2011*) and, therefore, how much replication there is in the system. Those examples where fornix lesions appear more disruptive than MB lesions (*Aggleton et al., 1995*; *Gaffan et al., 2001*) are consistent with the idea that the direct hippocampal-thalamic inputs are supportive, but there are some situations where anterior thalamic or MB lesion effects can be greater than those of fornix lesions (*Tonkiss and Rawlins, 1992*; *Vann et al., 2000*; *Jenkins et al., 2004*; *Aggleton et al., 2009*), so highlighting the potential functional importance of additional information streams, for example, from the limbic midbrain.

The present findings clearly suggest that the more crucial inputs for MB spatial function are from VTNg rather than the hippocampal formation. Questions immediately arise about the information provided by these tegmental nuclei and how might this relate to spatial memory. VTNg cells fire rhythmically and coherently with hippocampal theta (*Kocsis et al., 2001*), and theta-related cells in the medial mammillary nucleus have been linked to memory function (*Alonso and Llinas, 1992*; *Kocsis and Vertes, 1994*; *Bland et al., 1995*; *Kirk et al., 1996*). It is also the case that the VTNg have been linked to vigilance states (*Bassant and Poindessous-Jazat, 2001*), and so might additionally modulate hippocampal activity. The presumption, based on their dense interconnectivity, is that the VTNg lesion effects are primarily via the MBs. The VTNg does, however, innervate a number of other sites, including the lateral hypothalamus, preoptic area, medial septum, parts of the reticular pontine nucleus, median raphe nuclei, supramammillary nucleus and ventral tegmental area (*Petrovicky, 1973*; *Leichnetz et al., 1989*; *Hayakawa et al., 1993*). These other VTNg efferents appear light (see *Hayakawa and Zyo, 1984*), and in some instances not been confirmed (*Barone et al., 1981*; *Vertes, 1988*), a contribution from their disconnection must be considered, particularly because some of these other sites have hippocampal connections. However, lesions of these alternative sites rarely result in the pattern of spatial memory impairments seen following VTNg lesions (e.g., *Sarihi et al., 2000*; *Pan and McNaughton, 2002*). In addition, the lack of Fos changes in the supramammillary and septal nuclei following VTNg lesions provides some evidence that they are not having their effects via these routes, while the striking similarity of the VTNg and MTT lesion effects in the present study provides further support for the VTNg-MB pathway.

In rats, the MTT and VTNg lesion effects appear to be selective for spatial working memory and not due to non-specific effects such as loss of arousal or motivation. Both MTTx and VTNx groups were able to acquire a geometric discrimination task at the same rate, and to the same level, as the Sham group. This task may be particularly sensitive to lesions within the head-direction system (*Aggleton et al., 2009*; *Vann, 2011*). As the MTT lesions in the present study are most likely to disconnect the medial MB efferents, while leaving many lateral MB efferents largely intact (*Vann and Albasser, 2009*) this sparing provides further support for the dissociation of two memory systems within the MBs (*Allen and Hopkins, 1989*; *Vann and Aggleton, 2004*).

Clinical studies show that both the MBs and the MTT are important for human memory, and recollective memory in particular (*Dusoir et al., 1990*; *Van der Werf et al., 2000*; *Carlesimo et al., 2007*; *Tsivilis et al., 2008*; *Vann et al., 2009*). The situation regarding the human VTNg is far less clear. Indeed, for a number of years the very existence of this nucleus in humans had been in doubt (*Petrovicky, 1971*; *Hayakawa and Zyo, 1983*). However, it is now apparent that the structure and connections of the VTNg are remarkably similar across species (*Petrovicky, 1973*; *Irle and Markowitsch, 1982*; *Hayakawa and Zyo, 1984*; *Huang et al., 1992*; *Saunders et al., 2012*). To date, there is only one reported amnesic patient with restricted damage in the VTN area, but this study is limited in that only CT images were available (*Goldberg et al., 1981*).

The current findings support a novel model of limbic brain interactions. They show that the MBs have important roles in memory that can be independent of their hippocampal inputs; instead, the results highlight the VTNg. Not only do MTT and VTNg lesions result in strikingly similar spatial memory impairments but they also both result in widespread hypoactivity across a memory-related network. These findings highlight the importance of looking beyond the hippocampus to larger networks of brain regions that contribute to memory (*Vann and Albasser, 2011*).

## Materials and methods

### Subjects and surgery

Subjects were 40 male, pigmented rats (Dark *Agouti* strain; Harlan, Bicester, United Kingdom) weighing between 214 g and 245 g at the time of surgery. Animals were housed in pairs under diurnal light

conditions (14 hr light/10 hr dark) and testing was carried out during the light phase. Animals were given free access to water and a large cardboard tube and wooden chew-stick were available in the home-cage throughout. For all behavioral experiments other than the water-maze, where food was available ad libitum, the animals were placed on a food restricted diet where they were still able to gain weight; their weights did not fall below 85% of their equivalent free feeding weight. All experiments were carried out in accordance with UK Animals (Scientific Procedures) Act, 1986 and associated guidelines.

Prior to surgery, all animals were deeply anesthetized by intraperitoneal injection of sodium pento-barbital (60 mg/kg pentobarbital sodium salt; Sigma-Aldrich, United Kingdom). The 12 rats receiving Gudden's ventral tegmental nuclei lesions (VTNx) were then placed in a stereotaxic headholder (David Kopf Instruments, Tujunga, CA), with the nose bar at 0.0; the scalp was cut and retracted to expose the skull. The lesions were made by injecting 0.09M N-methyl-D-aspartate (NMDA; Sigma Chemical Company Ltd, United Kingdom), dissolved in phosphate buffer (pH 7.2). Injections (0.18 μl) were made in one site per hemisphere using a 1 μl syringe (Hamilton Bonaduz AG, Switzerland). The injection was made over 10 min and the needle was then left in situ for a further 10 min. The stereotaxic 'arm' was set at 20° from vertical so that the needle entered the same hemisphere for both injections. The stereotaxic co-ordinates of the lesion placements relative to bregma were anteroposterior (AP) −8.8, lateral (L) +2.6/+3.0 and the depth, from top of cortex, was −7.2 mm. For the mammillothalamic tract lesions (MTTx; n = 11) and descending postcommissural fornix lesions (PCFx; n = 12), the nose bar was set at +5.0. These lesions were made by radiofrequency using a thermocouple radiofrequency electrode (0.7 mm active tip length, 0.25 mm diameter; Diros Technology Inc., Toronto, Canada). For both lesion types, the electrode was lowered vertically and the tip temperature was raised to 60°C for 15 s using an OWL Universal RF System URF-3AP lesion maker (Diros Technology Inc., Toronto, Canada). The stereotaxic coordinates for the MTT lesions were: AP, −1.2; L, ±0.9 (both relative to bregma); and the depth, from top of cortex, was −6.9 mm. For the PCF lesions the coordinates were: AP, −0.2, L, ± 1.2, and the depth, from top of cortex, was −8.2 mm. There were three surgical control rats for each lesion type (Sham; n = 9) and for these surgeries the same procedures were used except the probe/needle was lowered to +1.0 mm above the lesion site; the temperature of the probe was not raised nor was any injection made.

During surgery, rats were maintained on oxygen and given an analgesic (Meloxicam; Boehringer Ingelheim, Rhein, Germany). At the completion of surgery, the skin was sutured and an antibiotic powder (Acramide; Dales Pharmaceuticals, Skipton, UK) was applied topically. Animals also received subcutaneous injections of 5 ml glucose saline. All animals recovered well following surgery. Behavioral testing began four weeks following the completion of surgery.

## Apparatus and behavioral training

### Experiment 1: reinforced spatial alternation in the T-maze

#### Apparatus

Two identical cross-mazes were used during the experiment (*Figure 2A*). When both mazes were in use, they were placed side-by-side, with the ends of the two arms touching each other. Details of the mazes and the room have been previously reported (*Vann, 2011*). Access to an arm could be prevented by placing an aluminum barrier at the entrance of that arm.

During the initial habituation and training in the 'light' stages (Stages 1 and 2), lighting was provided by two standard ceiling lights giving a mean light intensity of 140.8 lx (measured at the center of the two mazes). During training in the 'dark' stage (Stage 3), lighting was provided by a standard lamp, equipped with a 60W red light bulb (giving a mean light intensity of 1.4 lx at the center of the mazes), which was placed on the floor of the test room under the centre of the table.

#### Procedure

Prior to testing, animals were given 5 days of maze habituation. During each habituation session, reward pellets (45 mg; Noyes Purified Rodent Diet, United Kingdom) were placed in the maze and continuously replaced so that no arm was found to be empty on return. The start arm was separated from the choice arms so that the rats were allowed to run up and down the separate arms during habituation but not turn a corner. The rats were equally habituated to both mazes. Training on the forced-choice alternation rule commenced after the last day of habituation. The procedure used was the same as that previously used for rats with retrosplenial cortex lesions and lateral mammillary body lesions (*Pothuizen et al., 2009*, *2010*; *Vann, 2011*).

## Basic procedure for stages 1–3

Throughout the entire experiment, the rats received eight trials per daily session. Each trial consisted of a forced 'sample' run followed by a 'choice' run (*Figure 2A*). All testing was carried out in a cross-maze, which was modified by placing a barrier blocking access to the arm directly in line with the start arm (effectively transforming it into a T-maze).

During the forced sample run, one of the side arms of the maze was blocked by an aluminum barrier. After the rat turned into the preselected arm, it was allowed to eat the two reward pellets that had been previously placed in the food well. The rat was then picked up from the maze and immediately returned to the start arm, either in the same maze (Stage 1) or in the second maze (Stages 2–3)—see *Figure 2A*. Approximately 3 s after the end of the sample phase the choice phase began. The rat was allowed to run up the start arm and was now given a free choice between the left-turn and right-turn arms. The rat received two reward pellets only if it turned in the direction opposite to that in the sample run (i.e., non-matched). The arm (left or right) that the animal was 'forced' to go down during the sample stage varied pseudo-randomly. The rats were transported between the holding room and the experimental room in an opaque aluminum traveling box. Groups of either three or four rats were tested together, with each rat having one trial in turn, so that the inter-trial interval was approximately 3 min.

## Stage 1 (standard T-maze alternation; all cue types available)

In Stage 1 (six sessions collapsed into three, two-session blocks), both the sample and choice run were carried out in the same maze (*Figure 2A*). Both mazes were used equally and the use of each maze alternated between sessions. The maze not in use was removed from the experimental room. All sample runs started from the South. For half of the standard alternation trials the rat was forced to the West in the sample run, and rewarded for going into the east arm on the choice run (*Figure 2A*). For the remaining standard alternation trials the rat was forced to go East in the sample run, and rewarded for going West on the choice run. For these standard alternation trials all cue types could be used to support performance (i.e., visual allocentric, intra-maze, direction, and egocentric). Testing was counterbalanced with four trials per session, half to the West and half to the East.

## Stage 2 (visual allocentric, direction and egocentric cues available; intra-maze cues removed)

In Stage 2 (eight sessions, collapsed into four, two-session blocks), the sample and choice runs were conducted in different, adjacent mazes to prevent the use of intra-maze cues. The number of sample and choice runs was counterbalanced across the two mazes. The eight trials per session were equally divided across two distinct trial types (*Figure 2A*). For one trial type ('different place') the correct choice took the rat further away from the arm used in the sample run (*Figure 2A*). For the other trial type ('same place'), the rats were rewarded for choosing the arm leading to a goal that was located very close to the absolute position of the food well in the sample run. Thus for 'different place' trials all strategies except intra-maze cue use should show full transfer from Stage 1, while for 'same place' trials the visual allocentric cues could potentially interfere with arm choice. In contrast to Stage 1, equal numbers of sample runs (and choice runs) began from the North and the South start point (note, *Figure 1A* only shows trials starting from the South). However, to promote the use of direction (e.g., bearing) cues, the same start point (North or South) was always used for the sample and choice run on any given trial.

## Stage 3 (direction and egocentric cues available; visual allocentric and intra-maze maze cues removed)

Stage 3 (eight sessions, collapsed into four, two-session blocks) was identical to Stage 2 except testing was carried out in the dark to prevent the use of visual allocentric cues (*Figure 1A*).

## Experiment 2: radial-arm maze (acquisition and rotation)

Testing was carried out in an eight-arm radial maze. The maze consisted of an octagonal central platform (34 cm diameter) and eight equally spaced radial arms (87 cm long, 10 cm wide). The base of the central platform and the arms were made of wood and panels of clear Perspex (24 cm high) formed the walls of the arms. At the start of each arm was a clear Perspex guillotine door (12 cm high)

attached to a pulley. The maze was positioned in a room (255 × 330 × 260 cm) that contained salient visual cues such as geometric shapes and high contrast stimuli on the walls.

Pre-training for the radial-arm maze began 3 days after the completion of testing in the T-maze and involved two habituation sessions where the animals were allowed to explore the maze freely for 5 min with the guillotine doors raised and food pellets (45 mg; Noyes Purified Rodent Diet, United Kingdom) scattered down the arms. The animals were then trained on the standard radial-arm maze task (see below). A time limit of 10 min was placed on each session. Animals were tested until they had completed 15 sessions in Stage 1 and 6 sessions in Stage 2.

Stage 1 (sessions 1–15; collapsed into five, three-session blocks): At the start of a trial all eight arms were baited with two food pellets. The animal was allowed to make an arm choice and then return to the central platform. All the doors were closed for about 10 s before they were re-opened, permitting the animal to make another choice. This procedure continued until all eight arms had been visited or 10 min had elapsed. Only trials where animals made a minimum of eight arm choices were included in the analyses.

Stage 2 (sessions 16–21; collapsed into two, three-session blocks) tested for the possible use of intra-maze cues when performing the task. The start of the session was as before but after the animal had made four different arm choices it was contained in the center of the maze while the maze was rotated by 45° (clockwise/anticlockwise on alternate days). The remaining food pellets were moved so that they were still in the same allocentric room locations but the actual arms had changed; the experimenter pretended to bait each arm while moving the pellets so not to inadvertently cue the animal. The session then continued until all reward pellets had been retrieved.

## Experiment 3: geometric discrimination in the water-maze

The apparatus and general procedures are the same as previously described (e.g., *Vann, 2011*). Briefly, rats were given one session of pre-training in a circular pool where a landmark was attached to the platform and the platform was in a new position for each trial. The rats then received 12 sessions (analyzed as six, two-session blocks) with the rectangular insert. For this training there was no landmark and two platforms were present, one in each of the correct corners (see *Figure 4A*). The rectangular insert was rotated between trials so the platform positions did not remain fixed relative to external cues. The correct corners were balanced across groups. For all training the animals received four trials per day. The trials lasted a maximum of 120 s; if the rat had not located the platform by this time it was guided to the platform. At the end of the trial the rats remained on the platform for 30 s before being removed from the pool. A curtain was drawn closed around the pool for all pre-training and training. On day 13, the rats received three trials as normal but on the fourth trial the platforms were removed from the pool and the rats were allowed to swim for 60 s. The amount of time the animals spent in the correct and incorrect corners was noted.

## Experiment 4: *c-fos* expression following a radial-arm maze task

Approximately one week following completion of Experiment 3, rats were trained on a forced-choice radial-arm maze task for three days. The same maze and room were used as for Experiment 2 but the task differed; the experimenter controlled which arm the animals visited by using a pulley system to open the guillotine door at the start of the arm, that is, the animals were only allowed access to one arm at any one time. When all eight arms had been visited (i.e., the trial had been completed) the rat was placed in an aluminum traveling box for approximately 2 min while all arms were re-baited with sucrose pellets. Each session consisted of multiple trials in the radial-arm maze, one after the other, and lasted 20 min. Different randomized arm sequences were used on successive trials. On the final test day the animals performed the same task, in an identical radial-arm maze, but in a novel room (255 cm × 330 cm × 260 cm); the two radial-arm maze rooms were markedly different (i.e., size, shape, lighting) and contained distinct, salient visual cues such as geometric shapes and high-contrast stimuli on the walls. Animals were placed in a holding box in a quiet, dark room for 30 min before and 90 min after each radial-arm maze session. This time window was chosen as peak levels of the Fos protein are found approximately 90 min after activation (e.g., *Sharp et al., 1993*; *Chaudhuri et al., 2000*).

## **Histological and immunohistochemical procedures**

90 min after completing the final radial-arm maze session in the novel room, rats were deeply anesthetized with sodium pentobarbital (60 mg/kg; Euthatal, Rhone Merieux, United Kingdom) and transcardially

perfused with 0.1 M phosphate buffer saline (PBS) followed by 4% paraformaldehyde in 0.1 M PBS (PFA). The brains were removed and postfixed in PFA for 4 hr and then transferred to 25% sucrose overnight at room temperature with rotation. Sections were cut at 40 μm on a freezing microtome in the coronal plane. One series (one-in-four sections) was collected in PBS. Sections were processed for c-Fos immunostaining using c-Fos rabbit polyclonal antibody (1:3000; SC-52; Santa Cruz Biotechnology, Santa Cruz, CA). The methods have been described previously (*Vann and Albasser, 2009*).

A second, one-in-four series of sections was mounted onto gelatin-coated slides and stained with cresyl violet, a Nissl stain, for histological assessment. A further series (also one-in-four sections) was collected in PBS to be processed for either serotonin or acetylcholinesterase staining. Tissue from the VTNx rats and three Sham rats underwent acetylcholinesterase staining using the modified Koelle method as previously described (*Vann, 2009*).

## Image capturing and assessment

Images were captured using a Q Imaging MicroPublisher 3.3 RTV camera attached to a Zeiss Axiostar Plus microscope. Intensity measures for the acetylcholinesterase stained sections and estimates of IEG-positive cells were made using the public domain NIH Image program (developed at the US National Institutes of Health and available on the Internet at http://rsb.info.nih.gov/nih-image/). Counting and measuring procedures were carried out without knowledge of the group assignments.

## Laterodorsal tegmental nucleus integrity following VTN lesion

It is important that the VTNx lesions did not encroach into this structure. To assess the integrity of the cholinergic efferents from the laterodorsal tegmental nuclei, intensity measures were taken (following acetyl cholinergic staining) from the anteroventral thalamic nuclei and interpeduncular nucleus. Intensity measures were taken across four sections for the anteroventral and interpeduncular nuclei sections (i.e., eight total measures for the anteroventral thalamic nuclei for both hemispheres). Intensity measures were also taken from white matter adjacent to the nuclei of interest for each section: the stria medullaris of the thalamus for the anteroventral nucleus and the cerebral peduncle for the interpeduncular nucleus. The white matter value was then subtracted from the nucleus value, for each section separately, and the mean values calculated.

## Areas for c-Fos counting

The regions sampled for c-Fos counts were chosen either because they have previously been reported to be affected by mammillothalamic tract lesions (*Vann and Albasser, 2009*), because they receive efferents from the VTN or because they were 'control' regions to test the specificity of any findings. The hippocampus counts were taken at the level of 4.5 mm behind bregma (*Paxinos and Watson, 1997*); the counts included the dentate gyrus, CA3, and CA1, that is, not the subiculum complex. The retrosplenial cortex counts were taken at a level approximated 6 mm behind bregma and involved retrosplenial granular a and b (*Wyss and Van Groen, 1992*). Two frontal cortical regions were also examined: prelimbic cortex (PL) and infralimbic cortex (IL) taken at a level of 2.7 mm in front of bregma. The supramammillary nuclei (−4.3 mm from bregma), medial and lateral septum (+0.7 mm from bregma) were assessed as these regions have been reported to receive inputs from VTN. Finally, counts were taken from the primary somatosensory area (−2.5 mm from bregma) as any difference in this region might reflect abnormal sensorimotor patterns following surgery or non-specific lesion effects.

For all areas, other than the primary somatosensory cortex, the entire extent of the target region within the selected coronal sections was assessed. These counting procedures were not stereological, and so while providing information about relative numbers of cells they do not provide accurate counts of absolute cell numbers. All analyses were carried out using the raw cell counts. For all brain areas analyzed, counts were taken from at least four consecutive sections from each hemisphere and these counts were averaged to produce a mean.

## Statistical analyses

Group comparisons were carried out using parametric tests (ANOVA and *t*-test). The probability level of <0.05 was treated as significant. Significant group effects were further investigated using post-hoc comparisons (Tukey). When tests involved several sessions, repeated measure analyses were applied with appropriate corrections for any violations of sphericity using the Greenhouse-Geisser adjustment. Statistical analyses were carried out using PASW Statistics 18.0 running on Mac OS X 10.6.

## Acknowledgements

Thanks to John Aggleton, Moira Davies and Heather Phillips.

## Additional information

### Funding

| Funder | Grant reference number | Author |
| --- | --- | --- |
| Wellcome Trust | WT090954AIA | Seralynne D Vann |

The funder had no role in study design, data collection and interpretation, or the decision to submit the work for publication.

### Author contributions

SDV, Conception and design, Acquisition of data, Analysis and interpretation of data, Drafting or revising the article

### Ethics

Animal experimentation: All experiments were carried out in accordance with UK Animals (Scientific Procedures) Act, 1986 and associated guidelines. The work was carried out under UK Home Office Project license no 30/2761. All surgery was performed under sodium pentobarbital anaesthesia, and every effort was made to minimise suffering.

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
