## [Decision Letter]

Thank you for sending your work entitled “Dismantling Papez’ circuit for memory” for consideration at *eLife*. Your article has been favorably evaluated by a Senior editor, a Reviewing editor, and two reviewers (one of whom, Paul Dudchenko, wants to reveal his identity).

The Reviewing editor and the two reviewers discussed their comments before we reached this decision, and the Reviewing editor has assembled the following comments to help you prepare a revised submission.

This is a well-written manuscript that explores limbic functional connections by means of lesion and c-Fos studies of pathways involved in learning and memory in rats. For convenience, the authors use the rubric of the so-called Papez circuit to subsume under one umbrella circuits that focus on connections with the hippocampal formation.

The present study expands on several studies by these authors that are directed to determining the types and relative contributions of the components these behaviours (e.g., spatial memory, theta rhythms, and head direction) and the notion that there are two, or perhaps three, separate systems segregated with respect to the mammillary nuclei connections.

The authors show with very precise lesions that when the MB (reciprocal) connections with the ventral tegmental nucleus of Gudden (VTNg) or the connections of the ascending mammillothalamic tract (MMT) to the anterior thalamus are eliminated, large deficits in spatial memory occur while, in contrast, elimination of connections provided by the postcommissural fornix (PCF) have little effect.

The remarkable finding, in our view, is in the effects of the rotating the maze halfway through the animal's sampling of the different maze arms. This rotation causes the local cues of the maze to be dissociated from the extramaze cues available in the room. For the sham and PCFx lesioned animals, this rotation has no effect. This suggests, as one would expect, that extramaze cues are used in identifying which arms have been visited and which have not.

Please address the comments below in preparing a revised submission: 1) The authors show with very precise lesions that when the MB (reciprocal) connections with the ventral tegmental nucleus of Gudden (VTNg) or the connections of the ascending mammillothalamic tract (MMT) to the anterior thalamus are eliminated, large deficits in spatial memory occur while, in contrast, elimination of connections provided by the postcommissural fornix (PCF) have little effect. These are interesting and important results. In a more classical view of these structures, the lack of effect of PCF lesions can be considered surprising. What then is the role of the PCF? Some more detailed, explicit discussion of their ideas on this would be worthwhile in order to provide a clearer contrast with the dissociation of memory systems in the MB.

2) The effects on behaviour were correlated with reduced c-Fos expression in the retrosplenial cortex, the hippocampal formation, and the prelimbic cortex. It may well be that these reductions are the direct consequences of the lesions. On the other hand, the situation is likely to be rather complicated and other pathways could also be directly or indirectly involved.

3) With respect to the MB and the anatomical substrates of the results, we think more detailed anatomical analyses are warranted. For example, the authors note that the mammillary body is reduced in size (atrophied) as a result of PCFx and they attribute this to loss of white matter. However, it is difficult to determine this from Figure 1. Improved photomicrographs might help readers in making this assessment for themselves. The level of the MB selected does not seem to be at the middle level of the MB where medial, medial lateral, and lateral subnuclei are distinct, which would make evaluations easier. It may be as the authors state that a deafferentation (PCFx) large enough to cause MB atrophy has no (transneuronal) effect on the size of neurons but some measurements of the neurons might further substantiate this conclusion. Similarly, do MMT or VTNg lesions have no effect on MB neurons? It would be of interest to know if there are changes in the expression of c-Fos in the different MB subdivisions. Are there reasons why this was not done or not possible?

4) The hippocampus (e.g., hippocampus and dentate gyrus) are lumped together but no mention is made of, nor illustrations provided of, which neurons and layer are or are not c-Fos positive.

---

## [Author Response]

*1) The authors show with very precise lesions that when the MB (reciprocal) connections with the ventral tegmental nucleus of Gudden (VTNg) or the connections of the ascending mammillothalamic tract (MMT) to the anterior thalamus are eliminated, large deficits in spatial memory occur while, in contrast, elimination of connections provided by the postcommissural fornix (PCF) have little effect. These are interesting and important results. In a more classical view of these structures, the lack of effect of PCF lesions can be considered surprising. What then is the role of the PCF? Some more detailed, explicit discussion of their ideas on this would be worthwhile in order to provide a clearer contrast with the dissociation of memory systems in the MB*.

At present it is not clear what the role of the PCF pathway is. While it does not appear that this input is necessary for the tasks that have been used in the present study, it is possible that there are more sensitive tasks that would reveal effects and this needs further investigation. It is also possible that under normal circumstances this input stream does support aspects of these tasks, but these functions can be supported by other pathways when the PCF is lesioned, i.e., there is some redundancy in the system. This had previously been mentioned in the Discussion and has now been discussed further.

*2) The effects on behaviour were correlated with reduced c-Fos expression in the retrosplenial cortex, the hippocampal formation, and the prelimbic cortex. It may well be that these reductions are the direct consequences of the lesions. On the other hand, the situation is likely to be rather complicated and other pathways could also be directly or indirectly involved*.

In fact it is more than likely that these are indirect effects. None of the lesions with reduced c-Fos levels would be directly deafferented by the lesions in the present study. The changes, therefore, seem to reflect a dysfunction within a wider memory system. This has been re-iterated in the Discussion.

*3) With respect to the MB and the anatomical substrates of the results, we think more detailed anatomical analyses are warranted. For example, the authors note that the mammillary body is reduced in size (atrophied) as a result of PCFx and they attribute this to loss of white matter. However, it is difficult to determine this from Figure 1. Improved photomicrographs might help readers in making this assessment for themselves. The level of the MB selected does not seem to be at the middle level of the MB where medial, medial lateral, and lateral subnuclei are distinct, which would make evaluations easier. It may be as the authors state that a deafferentation (PCFx) large enough to cause MB atrophy has no (transneuronal) effect on the size of neurons but some measurements of the neurons might further substantiate this conclusion. Similarly, do MMT or VTNg lesions have no effect on MB neurons? It would be of interest to know if there are changes in the expression of c-Fos in the different MB subdivisions. Are there reasons why this was not done or not possible*?

Mammillary body area measurements have now been provided for the VTNx and MTTx lesion groups. The MBs are also significantly smaller in these groups compared to the Sham groups, although none of the lesion groups significantly differ from each other. Cell counts have now been provided and these are consistent with the MB atrophy in the PCFx lesion groups arising from loss of white matter and not cell loss.

An improved figure has now been provided that demonstrates the increased neuronal packing in the MBs of the PCFx lesion group compared to the Sham group.

Unfortunately in our laboratory we have never found reliable Fos staining in the mammillary bodies using selective antibodies, e.g., see page 984 of Vann, Brown and Aggleton, Neuroscience, 101:983-991, 2000. The counts are so low that a floor effect would be a major concern. To illustrate, please see the following photomicrograph of Fos staining in a control animal.

*4) The hippocampus (e.g., hippocampus and dentate gyrus) are lumped together but no mention is made of, nor illustrations provided of, which neurons and layer are or are not c-Fos positive*.

The difference in hippocampal counts seems to largely reflect changes in the CA1 subfield. The cell counts for the subfields CA1, CA3, and dentate gyrus are now provided. Fos-positive cells were only found in the stratum pyramidale of the CA fields and in the dentate gyrus Fos-positive cells were restricted to the stratum granulosum. This information has now been provided in the text.